# Combination of Radix Astragali and Safflower Promotes Angiogenesis in Rats with Ischemic Stroke via Silencing PTGS2

**DOI:** 10.3390/ijms24032126

**Published:** 2023-01-21

**Authors:** Shouchao Xu, Jiehong Yang, Haitong Wan, Li Yu, Yu He

**Affiliations:** 1School of Pharmaceutical Sciences, Zhejiang Chinese Medical University, Hangzhou 310053, China; 2School of Basic Medicine Sciences, Zhejiang Chinese Medical University, Hangzhou 310053, China; 3School of Life Sciences, Zhejiang Chinese Medical University, Hangzhou 310053, China

**Keywords:** Radix astragali and safflower, angiogenesis, ischemic stroke, ischemic penumbra, PTGS2

## Abstract

Promotion of angiogenesis and restoration of the blood flow in the ischemic penumbra is an effective treatment for patients with ischemic stroke (IS). Radix astragali-safflower (AS), a classic herbal pair for accelerating blood circulation and dispersing blood stasis, has been used for thousands of years to treat patients with IS in China. Even so, the mechanism of the treatment of IS by AS is still undecipherable. In the current study, network pharmacology was firstly employed to unveil the mechanism of AS in treating IS, which showed that AS might promote angiogenesis associated with PTGS2 silence. Middle cerebral artery occlusion/reperfusion (MCAO/R) model rats were then used as the experimental animals to verify the prediction result. The experimental results revealed that treatment with AS improved the cerebral infarct volume, neurological damage, and cerebral histopathological damage; inhibited cell apoptosis; increased the contents of PDGF-BB, EPO, and TGF-β1; and reduced the levels of PF4, Ang-2, and TIMP-1 in serum. Immunohistochemical staining demonstrated that the expression of PTGS2 was dramatically increased in the hippocampus and cerebral cortex of rats with MCAO/R, and this trend was reversed by the treatment of AS. Immunofluorescent staining expressed that AS reversed the down-regulation of VEGF and further promoted the expression of CD31, which indicated that AS promoted angiogenesis in MCAO/R rats. The abnormal protein or mRNA expression of PTGS2, PGI2, bFGF, TSP-1, and VEGF in the penumbra were transposed by AS or Celecoxib (an inhibitor of PTGS2). In conclusion, the protective mechanism of AS for IS promoted angiogenesis and was involved with PTGS2 silence.

## 1. Introduction

The prevalence of cerebrovascular diseases (CVDs) increases in the aging population [1]. Ischemic stroke (IS), characterized by the high disability and fatality rates, is one of the most serious CVDs [2]. IS is caused by cerebral artery occlusion, which ensues from an inadequate supply of blood and oxygen to the brain, and even local brain tissue necrosis [3]. The site of necrosis is called the cerebral infarction focus, which is constituted by the central necrotic area and surrounding ischemic penumbra [4]. The ischemic penumbra contains many dormant and non-necrotic neurons. A timely restoration of blood flow to the penumbra allows these neurons to survive; otherwise, these neurons would be apoptotic or necrotic [4,5]. Thus, protecting these reversible neurons in the penumbra has been the key strategy for the clinical treatment of patients with IS.

Promotion of angiogenesis in the penumbra and restoration of the cerebral blood flow perfusion are viewed as the core strategy to rescue the neurons in the penumbra. Reports have demonstrated that the survival rate of IS patients with high cerebrovascular density was higher than that of those with low cerebrovascular density [6]. One conceivable explanation for this is that angiogenesis recovers the supply of blood and oxygen in ischemic penumbra, activates the dormant neuron cells, and increases the survival rate of patients with IS. Thus, angiogenesis in the ischemic penumbra has been a new core strategy for treating patients with IS.

Angiogenesis is the development of new vessels from pre-existing vessels [7], which refers to the demolition of the vascular basement membrane; activation, proliferation, and migration of vascular endothelial cells; as well as the reconstruction of new blood vessels and vascular networks [8]. Angiogenesis is a complicated physiological process which is coordinated by various anti-angiogenesis and pro-angiogenesis cytokines. Vascular endothelial growth factor (VEGF), a universal expressed pro-angiogenic cytokine, has been considered as the most indispensable cytokine for the regulation of angiogenesis [6]. The VEGF receptor-2 (VEGFR2) is activated by endogenic VEGF to induce angiogenesis by several downstream signal pathways (such as PI3K/Akt and MEK/ERK pathways). Platelet-derived growth factor BB (PDGF-BB) and its receptor (PDGFRβ) mainly exist in brain pericytes [9], and it is located on extracellular matrix (ECM) or the surface of endothelial cells (ECs) to recruit brain pericytes into newly formed vessels during angiogenesis [10]. Furthermore, angiopoietin (Ang) involves the reestablishment and stabilization of newborn blood vessels by combining with Tie-2 receptors [9]. Thrombospondin (TSP) acts on its receptors to regulate cells’ mitosis and migration [11]. After IS, the homeostasis of these anti- and pro-angiogenesis factors is broken, resulting in the degeneration of the blood vascular system and deterioration of the prognosis of patients with IS.

Radix astragali-safflower (AS) is the classic herbal pair for accelerating blood circulation and dispersing blood stasis [12], and it appears in many traditional Chinese medicine prescriptions (such as Bu Yang Huan Wu Tang). Although AS has been used to treat IS for thousands of years, the mechanism of AS for treating IS is still unclear. Network pharmacology is a novel cross-discipline, which covers systems biology, bioinformatics, and network science. It clarifies the molecular mechanism between drugs and therapeutic targets for disease and reveals the systematic pharmacological mechanisms of drugs from the perspective of a system-level using a network [13]. To investigate the mechanism of AS against IS, network pharmacology was first employed, which showed that AS might promote angiogenesis associated with PTGS2 silence. Prostaglandin endoperoxide synthase 2 (PTGS2) is an important rate-limiting enzyme in prostaglandin biosynthesis. Previous research has expressed that PTGS2 involved the regulation of angiogenesis in plentiful cancer and tumor diseases [14,15,16]. Additionally, abnormal up-regulation of PTGS2 was observed in patients with IS [17]. Zhou et al. found that silencing of PTGS2 improved the injury induced by MCAO/R in mice [18], which suggested that PTGS2 was a potential target to treat IS. Consequently, whether AS promoted angiogenesis through inhibiting the expression of PTGS2 was investigated in this study.

## 2. Result

### 2.1. Chemical Components in AS Aqueous Extract

The data obtained by UPLC-Q-TOF-MS detection were processed by Unifi/Qi (2.1.1) software. The total ion chromatograms of AS aqueous extract in positive and negative ion modes are shown in Figure 1. According to the retention time, accurate mass, and MS/MS fragment information described in the literature, 54 compounds in positive and 49 compounds in negative mode were confirmed. Thirty-two compounds compliant with oral bioavailability (OB) ≥30% or drug-like properties (DL) ≥0.18 were considered as the bioactive constituents in AS aqueous extract for network pharmacology analysis. The name, retention time, molecule ID, molecular formula, and molecular weight of the thirty-two compounds are shown in Appendix A.

### 2.2. Network Pharmacology Analysis of AS against IS

To predict the potential targets of AS, 1083 AS−related targets (Appendix A) were obtained from SEA, Swiss Target Prediction, and TCMSP database. Analogously, 1621 IS−related targets (Appendix A) were obtained from OMIM, GeneCards, and DisGeNET database. There were 274 mutual targets (Appendix A) between AS and IS (Figure 2A), and these targets were considered as the potential therapeutic targets of AS against IS. The targets of each compound (AS1−AS32) are shown in Figure 2B. The Protein−Protein Interaction network (PPI, Figure 2C), with 224 nodes and 998 edges (interaction score > 0.9 and except disconnected targets), was sorted by centrality degree, and PTGS2 had the highest centrality degree among 274 mutual targets. The top 30 targets were selected for Kyoto encyclopedia of gene and genomes (KEGG) and Gene Ontology (GO) enrichment analysis. These targets were enriched in biological function related to cell migration and growth factor (Figure 2D). The KEGG pathway enrichment analysis (Figure 2E) expressed that the top 30 targets were associated with the VEGF signaling pathway. Consequently, we supposed that the protective effect of AS for IS was promoting angiogenesis through silencing PTGS2. 

### 2.3. AS Alleviated CIRI in MCAO/R Rats

TTC staining is a common method to evaluate the cerebral infarct volume. The results (Figure 3A,B) showed that the rat brains in the sham group were stained red, and that in the MCAO/R group, almost half were stained pale. The infarct volume in the MCAO/R group was 44.16 ± 3.58, suggesting that the rat MCAO/R model was successfully established. Compared to the MCAO/R group, the infarct volume was patently improved after administration of AS, Cel (an inhibitor of PTGS2), and Eda (a positive drug), and the infarct volume in the MCAO/R + AS, MCAO/R + Cel, MCAO/R + Cel + AS and MCAO/R + Eda groups were 21.60 ± 2.87, 25.40 ± 3.50, 13.20 ± 0.40, and 7.20 ± 1.47, respectively. There were no neurobehavioral deficits in the sham group (Figure 3C). Nevertheless, obvious forelimb flexure, turning to the side of paralysis, dropping from the balance beam, and reflection loss were observed in the MCAO/R group rats. Compared with the MCAO/R group, neurological injury was effectively ameliorated in the rats after administration of either AS and Cel or Eda. Treatment of AS or Cel alone did not improve the injury induced by MCAO/R. The possible explanation for this was that the data on neurobehavioral evaluation did not have Gaussian distribution.

Hematoxylin-eosin (HE) staining (Figure 3D) revealed that the pyramidal cells in the sham group were arranged neatly and tightly, nuclei were large and round, nucleoli were obvious, and cell morphology was intact, which verified that the cell morphology in the sham group was normal. Nevertheless, the pyramidal cells in the MCAO/R group were loosely arranged, cell nuclei appeared severely pyknotic or even disappeared, and large vacuolar structures appeared in the perinuclear region. These damaged alterations were improved after the administration of AS, Cel, or Eda. Compared to the MCAO/R group, clearer tissue structure was observed in the CA, Cel, or Eda−treated groups, indicating that the injury induced by MCAO/R was significantly alleviated. The cell morphology and structure in the AS + Cel group were comparable to those in the sham group and better than the Cel group, which suggested that AS alleviated cerebral ischemia reperfusion injury (CIRI).

### 2.4. AS Regulated the Expression of PDGF-BB, EPO, TGF-β, PF4, Ang-2, and TIMP-1 in Serum

Angiogenesis is a complex process that is regulated by multiple angiogenic factors. The downregulation of PDGF-BB, EPO, and TGF-β (angiogenic factors) and upregulation of PF4, Ang-2, and TIMP-1 (inhibition of angiogenic factors) in serum were observed in the MCAO/R group compared with the sham group (Figure 4A–F). Treatment with AS, Cel, or Eda showed the opposite effect on the levels of these cytokines. Compared with the MCAO/R group, the levels of PDGF-BB, EPO, and TGF-β were patently increased, and those of PF4, Ang-2, and TIMP were manifestly reduced in the AS, Cel, and Eda groups. 

### 2.5. AS Regulated the Expression of PTGS2 in Cerebral Cortex and Hippocampus

To study whether PTGS2 increased in rats with MCAO/R, immunohistochemical staining was used to quantitatively detect the expression of PTGS2. The analysis of PTGS2 in the cerebral cortex and in the penumbra is shown in Figure 5B,C. Compared with the sham group, the expression of PTGS2 was manifestly increased in the MCAO/R group, which confirmed that MCAO/R induced the abnormal increase in PTGS2. After treatment with AS, Cel, or Eda, the expression of PTGS2 was reversed, which suggested that AS reduced the expression of PTGS2. The same results are shown in Figure 5D,E. The expression of PTGS2 in hippocampal CA3 in the penumbra increased compared with the sham group, and AS, Cel, and Eda reduced it in the MCAO/R group. These results suggest an abnormal increase in PTGS2 in rats with MCAO/R, and AS reduced the expression of PTGS2 in the cerebral cortex as well as hippocampal CA3.

### 2.6. AS Promoted Angiogenesis in MCAO/R Rats

CD31 was a biomarker of angiogenesis, and the expression of CD31 in the ischemia penumbra was examined by immunofluorescence staining. The results (Figure 6A) expressed that the proportion of CD31-positive cells increased slightly in the MCAO/R group compared to the sham group, which indicated self-recovery after IS. AS or Cel treatment evidently increased the proportion of CD31-positive cells. The proportion of CD31-positive cells in the AS + Cel group was increased compared to that in the CA or Cel group. 

VEGF is a pro-angiogenic factor and plays an indispensable role in angiogenesis. Compared with the sham group, the proportion of VEGF-positive cells was patently reduced in the MCAO/R group, which indicated that MCAO/R inhibited the angiogenesis and retarded recovery after MCAO/R injury (Figure 6B). This finding notwithstanding, the proportion of VEGF-positive cells was increased after treatment with AS or Cel in comparison with the MCAO/R group. Furthermore, an increased proportion of VEGF-positive cells was observed in the AS + Cel group compared with the AS or Cel group. These results confirmed that AS treatment promoted angiogenesis in rats after MCAO/R.

### 2.7. The Effect of AS on the Protein and mRNA Expression of PTGS2, PGI2, TSP-1, Bax, Bcl-2, and VEGF

The results of network pharmacology analysis indicated that PTGS2 was the core target for AS against IS. PTGS2 is involved with angiogenesis in various tumor diseases. PGI2 is an important downstream target of PTGS2, and it involves endothelial cells sprouting and vascular permeability [19]. However, the role of PGI2 in patients with IS was still undecipherable. Thus, the expression of PGI2 was detected in this study. Compared with the sham group, the protein (Figure 7) and mRNA (Figure 8) expression of PTGS2 and PGI2 was increased in the MCAO/R group. AS and Cel reversed its abnormal expression. The up-regulation of TSP-1 and down-regulation of VEGF were observed in the MCAO/R group, and treatment with AS or Cel improved the abnormal expression of these protein and mRNA. At the same time, AS and Cel also reduced the apoptosis induced by IS. Consequently, AS promoted angiogenesis by repression of the PTGS2 gene.

## 3. Discussion

Three major findings concerning the protection of AS against IS were elucidated in this study. Firstly, AS administration alleviated the neurological damage after IS, as evidenced by reducing the infarct volume and the scores on motor, sensory, reflex and balance tests. Secondly, our results indicated that AS promoted angiogenesis in the penumbra by the regulation effect of anti- and pro-angiogenesis. Finally, the angiogenesis promotion effect of AS was reversed by Cel, an inhibitor of PTGS2, which indicated that PTGS2 was involved with angiogenesis after IS. These findings offer new insights into the likely regulatory mechanisms of AS.

IS, characterized by its high disability and fatality rates, is one of the most serious CVDs worldwide [2]. Recent research has explored the notion that promoting angiogenesis and restoring the blood flow in ischemic penumbra is an effective treatment route to reduce the death of neurons [20]. As a classic herbal pair for accelerating blood circulation and dispersing blood stasis, AS has been clinically applied for centuries. Network pharmacology analysis found that the core targets of AS were PTGS2, VEGFA, MMP9, MAPK1, and FGF2, and that the mechanisms of AS for treatment IS were associated with angiogenesis. 

Angiogenesis is the development of new vessels from pre-existing vessels [7], and it involves multiple molecular processes. Dissolution of the basement membrane marks the initiation of angiogenesis. Matrix metalloproteinases (MMPs) play an important role in degrading the ECM and the basement membrane. Nevertheless, MMPs are inhibited by a family of proteins known as tissue inhibitors of metalloproteinases (TIMPs) [21]. The expression of TIMP-1 significantly increased after acute cerebral ischemia and was involved in neurodegeneration [22]. The proliferation and migration of ECs depend on the intercoordination of anti-angiogenesis (PF4, TSP-1) and pro-angiogenesis (bFGF, TGF-β) factors. PDGF-BB and its receptor are mainly expressed in the brain, and their function is to recruit pericytes into newly formed blood vessels. Ang-2 mediates the stabilization and remodeling of newborn blood vessels by binding to the Tie-2 receptor. In the current study, the up-regulation of PDGF-BB, EPO, TGF-β, as well as the down-regulation of PF4, Ang-2, and TIMP-1, were observed after treatment with AS. Moreover, administration of AS decreased the infarct volume and alleviated the neurological damage induced by IS.

CD31 is a biomarker that is widely used to highlight the degree of neoangiogenesis [23]. Immunofluorescence staining analysis showed that CD31 had significantly lower expression in rats in the MCAO/R group. AS treatment obviously alleviated the decrease of CD31 in the brain tissue. VEGF is an indispensable pro-angiogenic factor with multiple biological functions, such as promotion of angiogenesis, neurogenesis neurite growth, and alleviation of cerebral edema [24]. Figure 8C,D revealed that the expression of VEGF patently reduced in the MCAO/R group, and administration of AS reversed the down-regulation of VEGF. These results indicated that the treatment of AS promoted angiogenesis in MCAO/R rats.

The molecular mechanism underlying the promotion of angiogenesis by AS was also investigated. Initially, the connection between PTGS2 and angiogenesis was major in cancer and tumors [14,15,16]. In recent years, increased expression of the PTGS2 protein in rats or mice with MCAO/R was observed [25], and its abnormal expression was associated with many pathogenic events that occurred in both the early and late stages of IS [26]. This evidence indicated that PTGS2 might be a potential target for IS treatment. The expression of PTGS2 was apparently elevated in rats with MCAO/R. Administration of AS or Cel reversed the abnormal elevation of PTGS2. Therefore, the angiogenesis promotion effect of AS was connected with PTGS2 silence.

## 4. Reagents and Methods

### 4.1. Reagents

Radix astragali (Batch number: 20210806) and safflower (Batch number: 20210818) were offered from Hangzhou Huadong Herbal Pieces Co., Ltd. (Hangzhou, China), and identified, respectively, as the root of *Astragalus membranaceus* (Fisch.) and the dried flower of *Carthamus tinctorius* L. by Professor Shuili Zhang from Zhejiang Chinese Medical University. Edaravone (Eda, Cat.HY-B0099/CS-1832) was offered by MedChemExpress Biotechnology Co., Ltd. (Shanghai, China). Celecoxib (Cel, batch number: C129279-5g) was offered by Shanghai Aladdin Biochemical Technology Co., Ltd. (Shanghai, China). The PDGF-BB, EPO, TGF-β, PF4, and Ang-2 enzyme-linked immunosorbent assay (ELISA) kits were provided by Jiangsu Meibiao Biotechnology Co., Ltd. (Nanjing, China). The TIMP-1 ELISA kit was provided by Henghui Biotechnology Co., Ltd. (Beijing, China). All other materials and reagents were supplied by Wuhan Servicebio Technology Co., Ltd. (Wuhai, China).

### 4.2. Preparation of AS Aqueous Extract

Radix astragali (~120 g) and safflower (~40 g) were weighted and extracted with 4.8 L of ultrapure water (weight (g): volume (mL) = 1:30) through heating reflux twice, for 30 min each time. The filtrate was merged and concentrated under reduced pressure. The AS powder was prepared by freeze-drying, and then dissolved with normal saline to prepare the sample at a concentration of 0.3 g/mL for later use.

### 4.3. Authentication Compounds in AS Aqueous Extract

To analyze the compounds in AS aqueous extract, a ultra-performance liquid chromatographer, equipped with high-resolution quadrupole and time-of-flight tandem MS (UPLC-Q-TOF-MS, Waters SYNAPT G2-Si, Waters, Milford, MA, USA) with a ACQUITY UPLC^®^HSS T3 column (2.1 × 150 mm, 1.8 μm, No.02483104918370), was used. The conditions in positive ion mode and negative ion mode (ESI^+^/ESI^−^) were the same, and they were as follows: ion spray voltage of 2500 eV, heated capillary temperature of 400 °C, sheath gas (nitrogen) of 50 psi, and auxiliary gas (nitrogen) of 50 psi. The elution mobile phase consisted of acetonitrile (B) and 0.1% formic acid aqueous solution (A). The gradient elution procedure was as follows: 0–10 min, 95→90% (A); 10–25 min, 90→80% (A); 25–55 min, 80→60% (A); 55–65 min, 60→5% (A). The elution flow rate was 0.3 mL/min.

### 4.4. Network Pharmacology Analysis

The potential medicative compounds in AS aqueous extract were determined according to the UPLC-Q-TOF-MS result. The criterion for selection compounds was OB ≥ 30% or DL ≥ 0.18 in TCMSP. Potential targets of main medicative compounds were searched from the Swiss Target Prediction (http://old.swisstargetprediction.ch/, accessed on 15 September 2022), TCMSP (https://old.tcmsp-e.com/tcmsp.php, accessed on15 September 2022) and SEA (https://sea.bkslab.org/, accessed on 15 September 2022) databases. These targets were converted to standard gene names using the Uniprot website. 

The disease targets of IS were collected from the OMIM (https://www.omim.org/, accessed on 15 September 2022), GeneCards (https://www.genecards.org/, accessed on 15 September 2022), and DisGeNet (https://www.disgenet.org/, accessed on 15 September 2022) databases. PPI network was performed by the STRING website and Cytoscape (3.8.2.0) software. GO and KEGG enrichment analysis of selected targets were performed by MetaScape and WebGestalt online tools, respectively.

### 4.5. Animals 

All adult male Sprague–Dawley rats (280–300 g) were provided by the Animal Center of Zhejiang Chinese Medical University (Laboratory Animal Certificate: SCXK 2017-0005). The rats were fed in specific pathogen-free grade conditions and supplied food and water ad libitum. The experimental procedures involving rats were implemented in accordance with the Institutional Animal Care and Use Committee of Zhejiang Chinese Medical University. The rules of 3R were followed during the whole experiment. The rats were fasted for 12 h before the experiment.

### 4.6. Establishment of Rat MCAO/R Model

The rat middle cerebral artery occlusion/reperfusion (MCAO/R) model was established using the suture-occluded method. Operation of the MCAO/R model was carried out as previously described [27]. A small longitudinal incision was made in the middle neck to expose the common carotid artery (CCA), external carotid artery (ECA), and internal carotid artery (ICA). A small hole at the 5–6 mm proximal end of the CCA bifurcation was made, and then a nylon suture was inserted through the incision to induce ischemic stroke in rats. After one hour of cerebral ischemia, the nylon suture was taken out for reperfusion. The operations in the sham group were the same, except that the nylon suture was not inserted.

### 4.7. Administration of Drugs

The rats were randomized into the following groups: a sham group, a MCAO/R group, a MCAO/R group with AS, a MCAO/R group with Cel, a MCAO/R group with Cel and AS, and a MCAO/R group with Eda. Cel was used as an inhibitor of PTGS2, and Eda was used as a positive drug. The rats in different drug treatment groups were treated with AS, Cel, and Eda by oral gavage once a day for 14 d, and the dosages were as follows: 3.6 g/kg, 10 mg/kg, and 10 mg/kg, respectively. The rats in the sham and MCAO/R groups were treated with the same volume of normal saline (Figure 9). The dosages of Cel and Eda were determined by reports from the literature [28,29]. The dosage of AS for rats was converted according to the human clinical dosage (40 g/70 kg). 

### 4.8. Evaluation of Neurobehavioral Deficits

Neurobehavioral deficits in rats were estimated by modified neurological severity scoring (mNSS) system in a double-blinded manner [30]. The mNSS system covered four aspects, including motor, sensory, reflex, and balance tests. The total score for neurobehavioral deficits was the sum of the scores for individual aspects, and ranged from 0 to 18. According to the score, the neurobehavioral deficits of rats were divided into four levels as follows: no functional damage, 1–6: mild damage; 7–12: moderate damage; 13–18: severe damage. Six independent experiments were performed.

### 4.9. Examination of Cerebral Infarct Volume

After administration of AS for 14 days, rats were sacrificed and the brains were cut into six two-millimeter-thick coronal slices. Then, 2,3,5-triphenyltetrazolium chloride (TTC) was used to incubate each slice at 37 °C for 15 min and protected from light. Finally, the stained sections were aligned from the smallest to the largest and photographed. After staining with TTC, normal brain tissue was red; however, infarcted brain tissue or ischemic brain tissue was pale. The infarct size for each slice was quantified using Image J software (1.4.3.67), and six rats per group were counted. Six independent experiments were performed. The formula for calculating the percentage of infarct volume in the entire brain was as follows:(1)$$The\;\mathrm{percentage}\;\mathrm{of}\;\mathrm{infarct}\;\mathrm{volume}\;\left(\%\right)=\frac{2\times \sum_{i=1}^{6}S_{i}^{infarct\;area}\;}{2\times \sum_{i=1}^{6}S_{i}^{whole\;area}\;}\times 100\%\;$$
where, $S_{i}^{infarct\;area}$ is the infarcted area of the *i*th slice; $S_{i}^{whole\;area}$ is the whole area of the *i*th slice; and 2 is the thickness of each slice.

### 4.10. Hematoxylin-Eosin (HE) Staining

The rats were sacrificed after AS administration for 14 days, and their brains were immersed in 4% paraformaldehyde at 4 °C for overnight. Then, the brain was dehydrated by gradient ethanol, embedded in paraffin, and cut into 4 μm coronal slices. The procedures for HE staining were as follows: the slices were dewaxed by xylene, rehydrated by rehydration, stained by hematoxylin, washed by tap water, differentiated by hydrochloric acid alcohol, and stained with eosin. Finally, the sections were sealed with neutral resin for observation using a microscope (ECLIPSE E100, NIKON, Tokyo, Japan) at 200× and 400× magnification. HE staining was observed six different fields per rat, and each group contained three separate rats. Three independent experiments were performed.

### 4.11. Detection of the Ang, PF4, EPO, TGF-β, PDGF-BB and TIMP Contents in Rat Serum

The whole blood of rats was obtained by cardiac puncture, and serum was prepared by centrifugation at 3500 rpm for 12 min. The contents of Ang, PF4, EPO, TGF-β, PDGF-BB, and TIMP in serum were detected by the corresponding ELISA kits. According to the instructions for each kit, each hole was detected at 450 nm using a microplate reader (MK-3, Thermo Fisher Scientific, Waltham, MA, USA) to obtain the optical density (OD) value. The contents of Ang, PF4, EPO, TGF-β, PDGF-BB, and TIMP in the serum of rats were calculated using standardized curves. Six independent experiments were performed to detect the contents of Ang, PF4, EPO, TGF-β, PDGF-BB, and TIMP.

### 4.12. Detection the mRNA Levels of PTGS2, PGI2, TSP-1, bFGF, VEGF, Bcl-2, and Bax Using PCR

The RNA was extracted from the homogenate of the penumbra by an RNA-extracting reagent and was reverse transcribed as cDNA in term of the instructions with 5X All-In-One RT MasterMix (Applied Biological Materials, British Columbia, Canada). PCR was performed to amplify sequences of PTGS2, PGI2, TSP-1, bFGF, VEGF, Bcl-2, and Bax, according to the instructions, with EvaGreen qPCR MasterMix-ROX (Applied Biological Materials, British Columbia, Canada) using the PCR Amplifier (Lightcycler 480Ⅱ, Roche, Basel, Switzerland). The mRNA levels were calculated by relative quantification (2^−ΔΔCt^), and GAPDH served as an internal reference gene. Three independent experiments were performed. The primers of PTGS2, PGI2, TSP-1, bFGF, VEGF, Bcl-2, and Bax were synthesized by Sangon Biotech Co., Ltd. (Shanghai, China), and the primer sequences were shown in Table 1.

### 4.13. Immunofluorescence Staining

The brains of rats was immersed in 10% paraformaldehyde for 24 h, and cut into 4 μm coronal slices after embedding with paraffin. The paraffin slices were parched at 60 °C for 1 h and dewaxed with graded ethanol and xylene. Normal goat serum was utilized as a blocking solution for 30 min at room temperature, and the primary anti-VEGF antibody (1:200, cat. no. GB13034, Wuhan Servicebio Technology Co., Ltd. Wuhan, China), as well as anti-CD31 (1:200, cat. no. GB12063, Wuhan Servicebio Technology Co., Ltd. Wuhan, China), were used to incubate the selections at 4 °C for 12 h. Then, the anti-rabbit secondary antibody (1:300, cat. no. GB21301, Wuhan Servicebio Technology Co., Ltd. Wuhan, China) was used to incubate the sections, and the sections were washed by PBS 3 times. Finally, the section was observed using an inverted fluorescence microscope (Ts2-FC, NIKON, Tokyo, Japan) at 400× magnification. The numbers of positive cells in predefined areas were quantified using Image J software. Six different fields per rat were observed, and each group consisted of three rats. All counts were conducted by blinded observers, and three independent experiments were performed.

### 4.14. Immumohistochemicalstaining

The brains were immersed in 10% paraformaldehyde for 24 h to obtain 4 μm paraffin sections. The paraffin sections were dewaxed with xylene and gradient ethanol (ethanol, 85% ethanol, 75% ethanol and distilled water). The sections were incubated for 15 min in citrate antigen repair buffer (pH 6.0) and washed by PBS 3 times (10 min each time). After incubation with 3% BSA for 30 min, the primary anti-PTGS2 antibody (1:200, Servicebio, Wuhai, China) was added and the sections were incubated for 12 h overnight at 4 °C. Then, the sections were incubated with anti-rabbit secondary antibody (1:1200, Servicebio, Wuhai, China) for 50 min. Finally, the sections were stained with hematoxylin and sealed with neutral resin for observation using a microscope (Ts2-FC, NIKON, Tokyo, Japan) at 400× magnification. The numbers of positive cells in the predefined areas were quantified using Image J software. Each group contained three rats, and cells in six different fields per rat were counted. All counts were conducted by blinded observers, and three independent experiments were performed.

### 4.15. Western Blotting

After 14 administrations of AS, the rats were sacrificed and the brain tissue was removed. Afterward, 1 mL of lysis buffer was added to ischemic brain homogenate to obtain the total protein by centrifuging at 12,000 r/min for 5 min at 4 °C, and a bicinchoninic acid (BCA) kit was utilized to measure the concentration of the protein. The proteins were then separated by 10% SDS-PAGE gel and transferred to a PVDF membrane, which was incubated for 2 h with blocking solution containing 5% skimmed milk. After blocking solution incubation, the membrane was incubated again with diluted primary antibodies on a shaker at 4 °C overnight, then the secondary antibodies diluted in a 5% skimmed milk blocking solution were used to incubate the membrane on a shaker for 2 h at room temperature. Finally, the membrane was stained with electrochemiluminescence solution for observation. Images were acquired with a ChemiDoc imaging system (XRS + 1708265, Bio-Rad, Shanghai, China). The gray value of each strip was calculated using Gray analysis software (alphaEaseFC, Alpha Innotech, CA, USA). The relative expression of the target strip was expressed as the ratio of the gray value of the strip to the gray value of the internal reference strip. All of the above experiments were performed three times.

### 4.16. Statistical Analysis

The data were assessed with normality analysis and homogeneous analysis of variance. After normality and homogeneous of variance were confirmed, the data were analyzed by one-way analysis of variance (ANOVA) and the Student–Newman–Keuls (SNK) test to assess whether there were statistic differences between different groups; otherwise, a non-parametric test was executed to analyze the data. All data were presented as mean ± standard deviation (SD) and analyzed using SPSS19.0 software. A *p*-value less than 0.05 was considered statistically significant.

## 5. Conclusions

Taken together, our findings proved that AS reduced the infarct volume, alleviated neurological damage, regulated anti-angiogenesis (PF4, Ang-2 and TSP-1) and pro-angiogenesis (PDGF-BB, TGF-β and EPO) factors in serum, and promoted the expression of CD31 and VEGF. The abnormal protein or mRNA expression of PTGS2, PGI2, bFGF, TSP-1, and VEGF in the penumbra were transposed by AS. Consequently, the protective mechanism of AS for IS was promotion of angiogenesis, which was involved with PTGS2 silence.

## Figures and Tables

**Figure 1 ijms-24-02126-f001:**
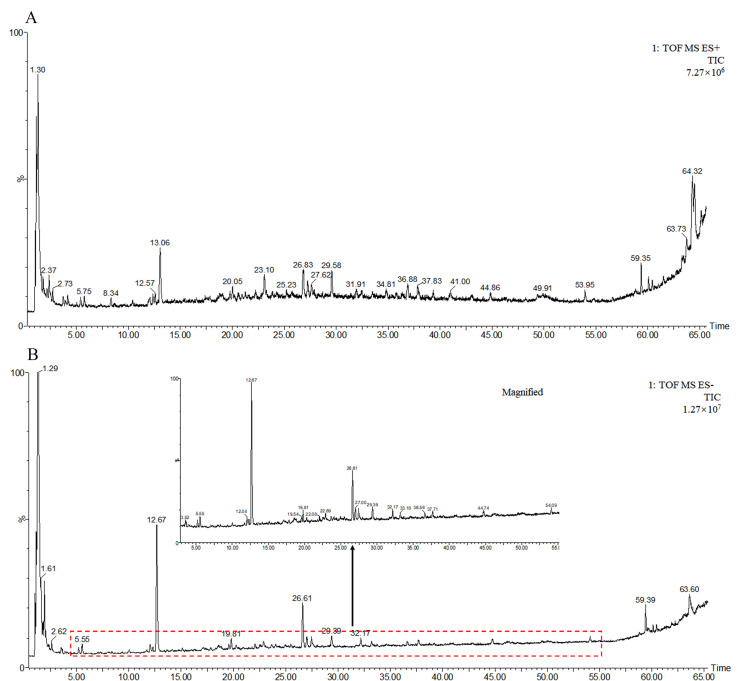
The total ion chromatograms of AS aqueous extract in positive (**A**) and negative (**B**) ion modes.

**Figure 2 ijms-24-02126-f002:**
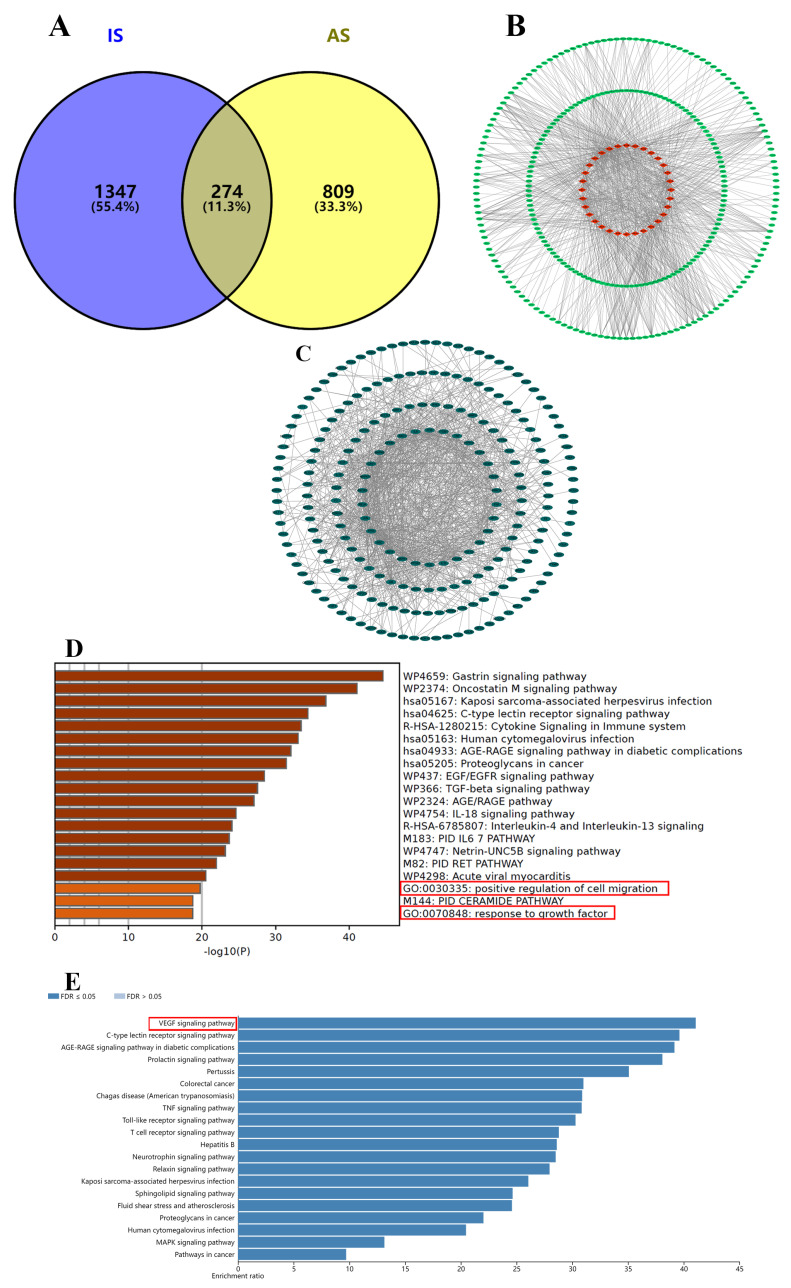
Network pharmacology analysis of AS. (**A**) Venn diagram of AS−related targets and IS−related targets. (**B**) Ingredient−target network of AS against IS. (**C**) The PPI network of 274 mutual targets. (**D**) The top 20 GO terms of core targets. (**E**) The top 20 KEGG pathways of core targets.

**Figure 3 ijms-24-02126-f003:**
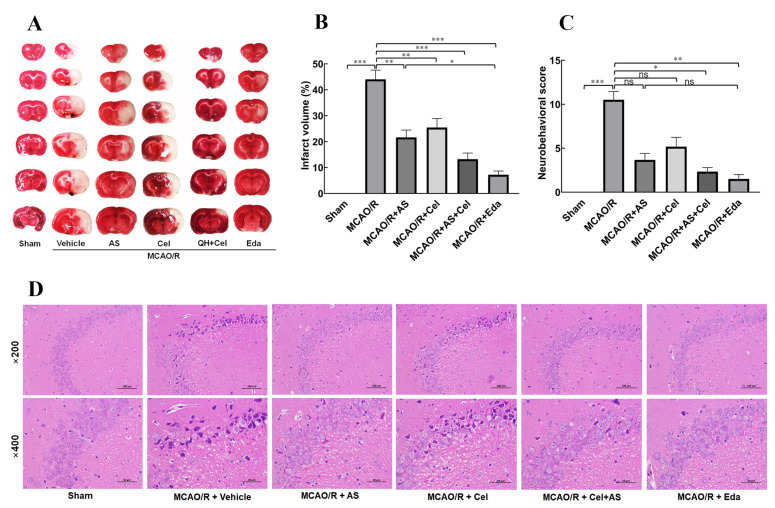
AS treatment improved CIRI in MCAO/R rats. (**A**) TTC staining was performed to evaluate infarct volume. (**B**) The infarct volumes of different groups were calculated. (n = 6 rats for each group) (**C**) Neurobehavioral deficits were detected by mNSS scoring at 14 d after reperfusion. (n = 6 rats for each group) (**D**) HE staining revealed the pathological morphology. (n = 3 rats for each group). Data are expressed as the mean ± SD, * *p* < 0.05; ** *p* < 0.01; *** *p* < 0.001; ns stands for no statistical significance.

**Figure 4 ijms-24-02126-f004:**
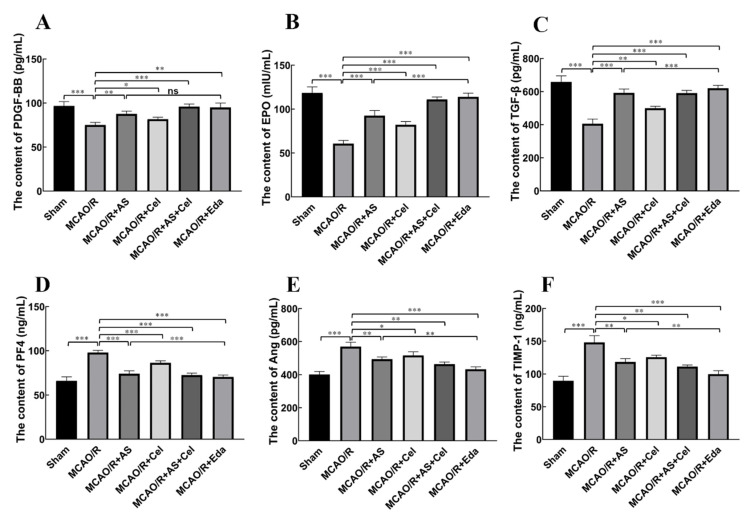
AS regulated the expression of PDGF-BB (**A**), EPO (**B**), TGF-β (**C**), PF4 (**D**), Ang-2 (**E**), and TIMP-1 (**F**) (n = 6 rats for each group). Data are expressed as the mean ± SD, * *p* < 0.05; ** *p* < 0.01; *** *p* < 0.001; ns stands for no statistical significance.

**Figure 5 ijms-24-02126-f005:**
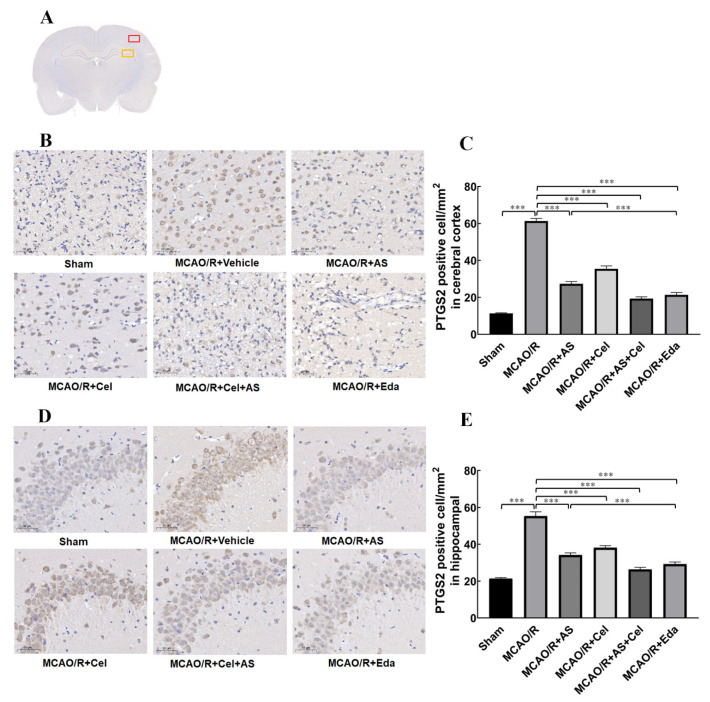
AS treatment inhibited the expression of PTGS2. (**A**) The observation areas in the cortex (red) and hippocampus (yellow). (**B**) The immunohistochemical staining (400×) of PTGS2 in the cortex and (**C**) quantitative analysis. (**D**) The immunohistochemical staining (magnification was 400×) of PTGS2 in the hippocampus and (**E**) quantitative analysis. (n = 3 rats for each group). Data are expressed as the mean ± SD, *** *p* < 0.001. ns stands for no statistical significance.

**Figure 6 ijms-24-02126-f006:**
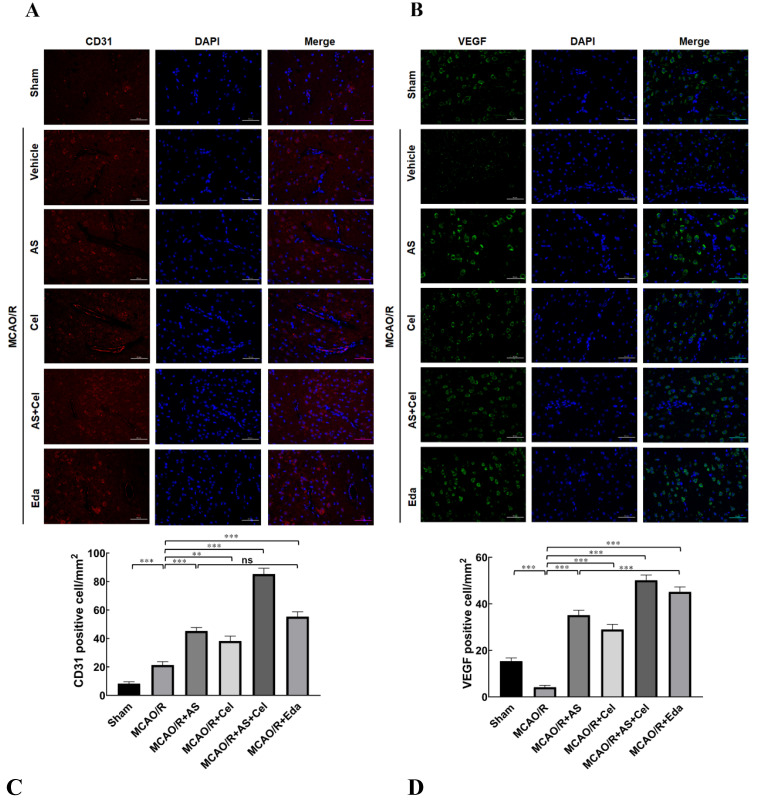
AS treatment promoted angiogenesis. (**A**) The immunofluorescent staining (magnification was 400×) of CD31 and (**B**) quantitative analysis. (**C**) The immunofluorescent staining (magnification was 400×) of VEGF and (**D**) quantitative analysis. Red fluorescence indicates CD31 signal, and green fluorescence indicates VEGF signal. (n = 3 rats for each group). Data are expressed as the mean ± SD, ** *p* < 0.01, *** *p* < 0.001; ns stands for no statistical significance.

**Figure 7 ijms-24-02126-f007:**
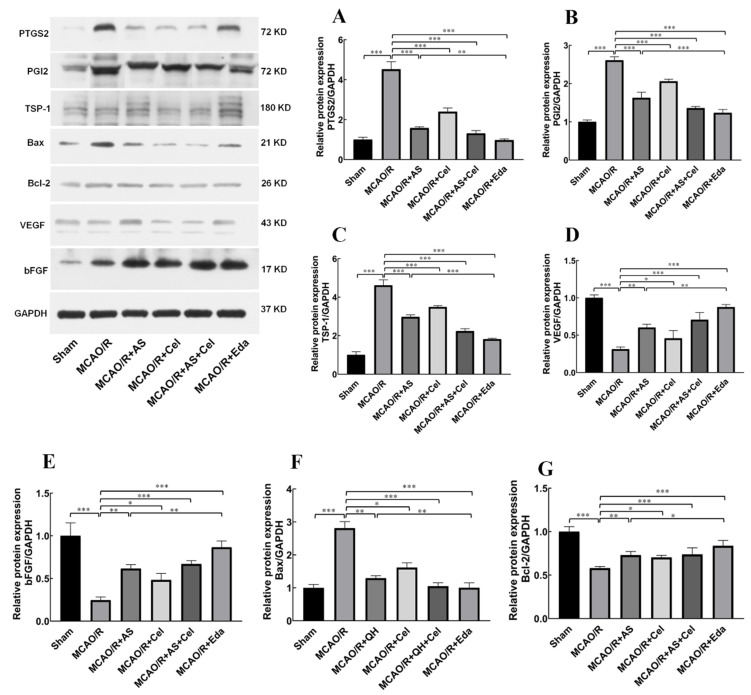
AS treatment promoted angiogenesis by inhibiting PTGS2. The protein expression of PTGS2 (**A**), PGI2 (**B**), TSP-1 (**C**), Bax (**D**), Bcl-2 (**E**), VEGF (**F**), and bFGF (**G**) was detected by Western blot. (n = 3 rats for each group). Data are expressed as the mean ± SD, * *p* < 0.05; ** *p* < 0.01; *** *p* < 0.001.

**Figure 8 ijms-24-02126-f008:**
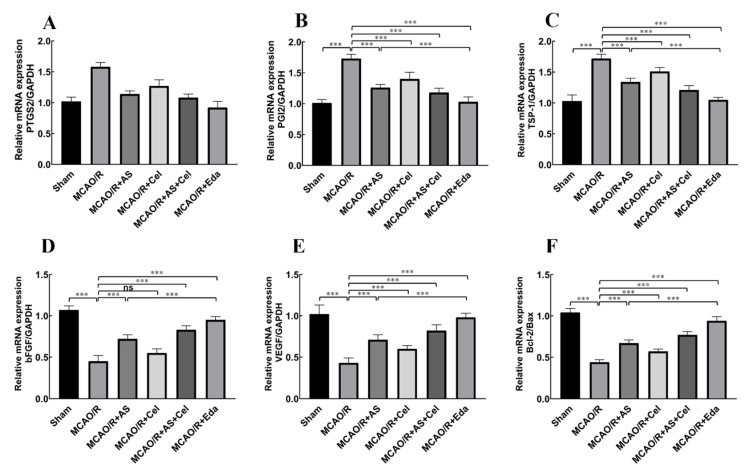
AS treatment promoted angiogenesis by inhibiting PTGS2. The mRNA expression of PTGS2 (**A**), PGI2 (**B**), TSP-1 (**C**), Bax (**D**), Bcl-2 (**E**), VEGF (**F**), and bFGF was detected by PCR. (n = 3 rats for each group). Data are expressed as the mean ± SD, *** *p* < 0.001; ns stands for no statistical significance.

**Figure 9 ijms-24-02126-f009:**
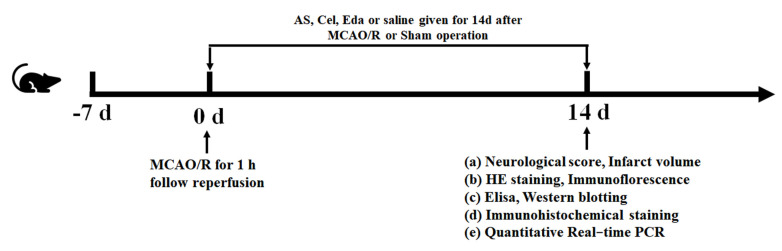
The schedule and design of the experiment.

**Table 1 ijms-24-02126-t001:** The primer sequences for RT-qPCR.

Gene	Sequence
PTGS2	Forward: 5′- CTCAGCCATGCAGCAAATCC-3′
	Reverse: 5′-GGGTGGGCTTCAGCAGTAAT-3′
PGI2	Forward: 5′-CTCTGGGCAACCGAGGAAAT-3′
	Reverse: 5′-GTTGTGGCGGATGGAGTTCTT-3′
TSP-1	Forward: 5′-GATGTCCGACTTATTCGAGAGC-3′
	Reverse: 5′-TTGAGCTGTAAGCGCCTTCTA-3′
bFGF	Forward: 5′-GGGGACTTGGTTGCCTTTT-3′
	Reverse: 5′-CAGCCATCGCAGATCACATT-3′
VEGF	Forward: 5′-CTGGAGAAACCTGCCAAGTATG-3′
	Reverse: 5′-GGTGGAAGAATGGGAGTTGCT-3′
GAPDH	Forward: 5′-CTGGAGAAACCTGCCAAGTATG-3′
	Reverse: 5′-GGTGGAAGAATGGGAGTTGCT-3′

## Data Availability

The data presented in this study are available on request from the corresponding author.

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
