# Peer review of "Combination of Radix Astragali and Safflower Promotes Angiogenesis in Rats with Ischemic Stroke via Silencing PTGS2"

_ijms, 2023, doi:10.3390/ijms24032126_

Round 1

Reviewer 1 Report

The research is very good and with high impact on the treatment of one of the most dramatic pathologies of our days. 

Reviewer 2 Report

This is a potentially significant study to identify mechanisms by which traditional Chinese medicine might impact the neurological damage associated with ischemic stroke. However, there are significant concerns: (1) convincing data that angiogenesis is enhanced is lacking; (2) Methods for image analysis are lacking for HE and IF staining as well as western blot analysis. Additional information is required for data analysis: the definition and value of “n” and the number of independent experiment performed to generate and analyze data.

Specific  concerns.

1.     Figure 6 is critical to the main conclusion of the study that the combination of Radix astragali and Safflower promotes angiogenesis. However, staining for CD31 is not detectable in images provided.

2.     No information is provided as to the identity of the thirty-two compounds chosen for analysis.

3.     Links to the data sets of AS and IS should be provided

4.     The identity of the 274 genes targeted by both AS and IS should be provided.

5.     Rationale for conditions of Cel (celecoxib) and Eda (edaravone) should be included in the description of experiments in Figure 3. Rationale for measuring PTGS2 and PG12. For example are these shared gene targets between AS and IS?

6.     Image analysis: No information was provided indicating how staining (IF and HE) was quantified. For example -how many images were analyzed per animal and how many animals were used per conditions? Was any image analysis software used for the analysis?

7.     Data analysis: (1) Figures 3-8. The authors need to (a) define “n” and (b) what “n” equals and (c) how many independent experiments were performed. (2) In the Figure legends the authors indicate that the data is presented as the mean +/- the SEM; however in the methods the authors indicated that all data was presented as the mean +/- SD. Which is it?

8.      Not all abbreviations used in the manuscript are included in the list of abbreviations. For example PPI, PTGS2, PF4, PGI2.

Round 2

Reviewer 2 Report

The staining for CD31 should be repeated. It is not clear what antibody was used. Most researches use an anti-CD31 antibody for BD PharMingen at 1:50 dilution. The authors used an antibody to CD31  at 1:200.  The staining is too weak.
